# Chitosan Coating Incorporated with Carvacrol Improves Postharvest Guava (*Psidium guajava*) Quality

Chang Shu [1,2], Beatrice Kim-Lee [1] and Xiuxiu Sun [1,*]

1   United States Department of Agriculture, Agricultural Research Service, Daniel K. Inouye U.S. Pacific Basin Agricultural Research Center, 64 Nowelo Street, Hilo, HI 96720, USA; chang.shu@usda.gov (C.S.); bkimlee311@gmail.com (B.K.-L.)
2   Oak Ridge Institute for Science and Education, 1299 Bethel Valley Road, Oak Ridge, TN 37830, USA
*   Correspondence: xiuxiu.sun@usda.gov; Tel.: +1-808-959-4307

**Abstract:** Guava (*Psidium guajava* L.) is an important economic crop grown widely in tropical and subtropical regions. Guava exhibits fast ripening and senescence as a climacteric fruit, causing a short shelf life and quality deterioration. Chitosan–essential oil nanoemulsions can be an edible coating used to improve postharvest quality attributes. In this study, chitosan was mixed with carvacrol to generate a nano-emulsoid solution containing 0.1 and 0.2% ($v/v$) carvacrol, using a sonic dismembrator. Guava fruit were coated with the above emulsion and postharvest quality parameters were measured during storage at $20 \pm 1$ °C and RH = $80 \pm 5\%$ for 8 days. The result illustrated that the particle size of the chitosan–carvacrol emulsions was nanoscale, and their high stability was demonstrated by the zeta potential and polydispersity index. Chitosan coating (2%, $w/v$, 310–375 kDa) containing 0.2% ($v/v$) carvacrol maintained postharvest quality compared to chitosan alone, with higher firmness, soluble solid content, total acid, and total phenol content, and lower weight loss and pericarp browning. The collective data were further verified by principal component analysis. A chitosan coating containing carvacrol can reduce postharvest losses. It can be applied as an effective strategy to improve postharvest fruit quality.

**Keywords:** edible coating; chitosan; carvacrol; guava; postharvest quality

## 1. Introduction

Guava fruit (*Psidium guajava* L.) is a significant economic crop cultivated extensively in tropical and subtropical regions [1]. It is known for its delectable taste and nutritional richness, and recent research also reveals its medical values [2]. However, guava is a climacteric fruit that undergoes rapid ripening and senescence, resulting in a short storage period and shelf life along with sensitivity to pathogen infection, leading to considerable postharvest losses [3]. Also, guava is sensitive to cold temperatures, and storage at temperatures below 10 °C may result in severe chilling injury symptoms [4]. These all significantly limit the production and commerciality of guava with postharvest losses as high as 10–24% [5]. Therefore, it is urgent to explore feasible methods to reduce postharvest losses and improve the quality of guava.

Researchers and industry fields have explored various postharvest technologies and treatments, including 1-methylcyclopropene [6], novel cold storage technology [7], ionizing radiation [8], hot water treatment [9], modified atmosphere packaging [10], melatonin treatment [11], and edible coatings [3]. These approaches aim to extend storage time and shelf life, inhibit decay, and maintain the nutritional quality of guava fruit. Among them, edible coatings are recognized as promising strategies. Edible coatings are mainly made from food-grade biopolymers, including polysaccharides, lipids, proteins, or a combination thereof [12]. They form a physical barrier on the surface of the fruit, directly prevent the invasion of pathogens, and regulate the atmosphere exchange of the fruit, reducing the postharvest respiration rate and consequent weight loss [13]. Since edible coatings are sold

and consumed as part of the fruit, current research primarily focuses on the use of generally recognized as safe (GRAS) substances to ensure their non-toxicity and food safety [14]. The incorporation of functional ingredients (such as antimicrobial and antioxidant agents) into the coating matrix has been demonstrated to improve its physicochemical properties [15]. Given the short shelf life of guava, where the quality rapidly deteriorates upon ripening, the application of edible coatings holds significant potential.

Chitosan, a natural polysaccharide derived from chitin, has garnered significant interest due to its desirable properties, including biodegradability, biocompatibility, antimicrobial properties, and film-forming ability [16]. Notably, the U.S. Food and Drug Administration (FDA) has recognized chitosan as a GRAS polymer [17]. These unique characteristics position chitosan as an excellent candidate for developing edible coatings to preserve fresh produce. Numerous studies have demonstrated the effectiveness of chitosan coatings in reducing respiration rates and microbial decay, delaying senescence, and maintaining postharvest fruit quality attributes [18,19]. Carvacrol (5-isopropyl-2-methyl phenol), a monoterpene compound mainly extracted from oregano and thyme, has received considerable attention due to its broad-spectrum antimicrobial activity and GRAS certification and may act as a novel bio-preservative [20]. Incorporating natural essential oils into polysaccharide-based coatings provides an additional protective barrier to control the release rate and reduce the volatile/oxidation loss of essential oil [21] while simultaneously contributing to better barrier and mechanical properties of the coating [15]. Furthermore, by reducing the particle size of incorporated essential oils to the nanoscale through the emulsion, the specific surface area of oil droplets is significantly increased [22]. This results in improved stability as they become more uniformly distributed within the film-forming solution, thereby enhancing the utilization efficiency of essential oil [23]. Even though the use of chitosan coatings loaded with nanoscale essential oil has shown promising results in extending the shelf life of various fruits, their application on guava still needs to be further studied.

This study aimed to prepare a chitosan coating containing nanoscale carvacrol and apply it as an edible coating on guava fruit. The coating solution was characterized, and the fruit's shelf-life and quality attributes were studied to provide an effective strategy for improving postharvest guava quality.

## 2. Materials and Methods

### 2.1. Reagents and Fruit Materials

Chitosan and carvacrol were both purchased from Sigma-Aldrich (St. Louis, MO, USA). The molecular weight of chitosan is between 310,000 and 375,000 Daltons with a deacetylation degree over 75%. The purity of carvacrol is 99% and it is food grade. Tween 80 was purchased from Research Products International Corporation (Mt. Prospect, IL, USA). Analytical-grade acetic acid was purchased from Fisher Chemical (Fair Lawn, NJ, USA), with a purity $\geq$ 99.7%. Other chemicals not mentioned were of analytical grade.

Guava fruit (*Psidium guajava* L.) was harvested about 20 to 28 weeks after flowering and pollination in Hilo, Hawaii, in October 2022. The fruits were uniform in size (about $130 \pm 15$ g per fruit) and free of any mechanical damage or diseases. The fruit surface was washed with deionized water and dried at room temperature for later use.

### 2.2. Preparation of Coating Solutions

Preparation of coating-forming solutions referred to a previous study with some modifications [24]. Chitosan was dissolved in distilled water containing 1% (*v/v*) acetic acid and 0.15% (*v/v*) Tween® 80 to obtain a 2% (*w/v*) chitosan solution. Carvacrol was then mixed into the chitosan solution to prepare 0.1% and 0.2% (*v/v*) carvacrol coating solutions. The solution was stirred at 700 rpm for 40 min to let carvacrol distributed evenly, then the mixture was ultrasonically homogenized at 20 kHz for 10 min (Model 705, Fisherbrand, Waltham, MA, USA) to generate a nano-emulsion as the coating-forming solution. The coating solution was kept at room temperature until further usage.

### 2.3. Coating Treatment and Storage Condition

The coating-forming solution was manually spread on the fruit surface. Each fruit was coated with 1.0 mL of the above solutions; to ensure consistent coating on each fruit, excess solution flowed down from the fruit surface. The coated fruits were allowed to dry naturally at room temperature until the surface was completely dry and then stored at $20 \pm 1\ ^\circ$C, RH = $80 \pm 5$% for 8 days to simulate shelf life. Fruit was sampled initially and at a 2-day interval during storage for quality evaluation. The whole experimental design is shown in Figure 1.

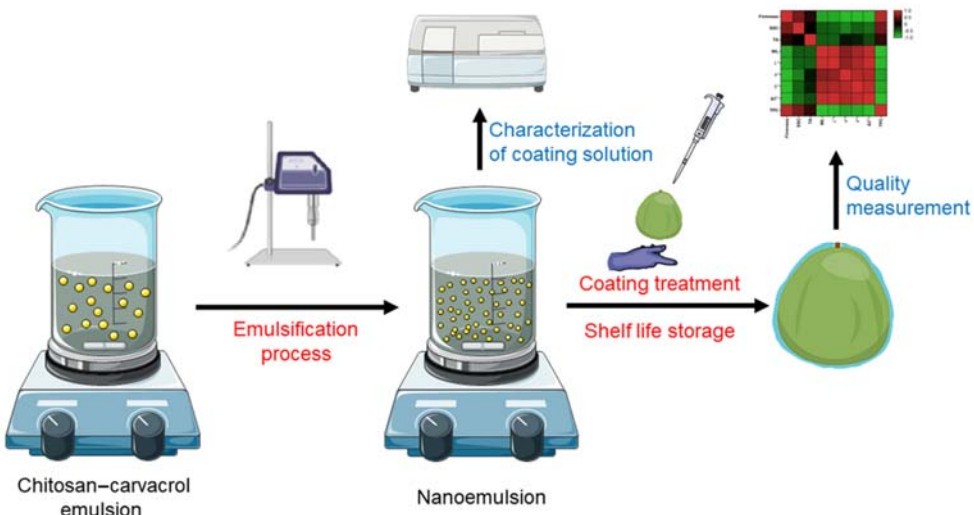

**Figure 1.** Schematic diagram of the experiment design.

### 2.4. Characterization of Coating Solution

The particle size, polydispersity index (PDI), and zeta potential of the samples were measured with the dynamic light scattering method using a Zetasizer analyzer (Zetasizer Ultrablue, Malvern Instruments Ltd., Worcestershire, UK). The data were analyzed by the XS Xplorer 3.2.0.84 software (Malvern Instruments Ltd., Worcestershire, UK).

### 2.5. Color Parameters

Peel surface color was measured on three fruits, and each fruit was measured at two opposite sites on the equator using a Minolta chromameter (model CR-300, Minolta Corp., Ramsey, NJ, USA) and recorded as CIE (International Commission on Illumination) $L^*$, $a^*$, and $b^*$. $\Delta E^*$ was calculated using the following Formula (1) to represent the total color difference when compared to the initial status.

$$\Delta E = \sqrt{(a_0 - a_1)^2 + (b_0 - b_1)^2 + (L_0 - L_1)^2} \tag{1}$$

### 2.6. Firmness

Fruit was peeled at two opposite sides of the fruit equatorial, and the tissue 30 mm under the epidermis part was measured. Pulp firmness was measured using a texture analyzer (Model Chatillon LTCM-100, AMETEK, Inc., Berwyn, PA, USA) equipped with a 60 mm diameter probe. The probe punctured the pulp 1.0 cm at a speed of 25.4 cm min$^{-1}$. The results were expressed in Newtons (N). In each replicate for each time point, a total of three fruits were randomly measured.

### 2.7. Total Soluble Solid Content and Titratable Acidity

Fresh pulp tissue was ground, and the juice was filtered through two layers of medical gauze. The total soluble solid content (SSC) was determined using a digital refractometer (PAL-3, ATAGO U.S.A., Inc., Bellevue, WA, USA), and the result was recorded as a percent-

age. The titratable acidity (TA) was measured with an acidity meter (GMK-835F, ATAGO U.S.A., Inc., Bellevue, WA, USA), which measures the total amount of hydrogen ions, and the result was expressed as a percentage.

### 2.8. Weight Loss

Weight loss was calculated as a percentage using the following Equation (2). The weight loss at each time point was fitted to linear models to estimate the daily weight loss.

$$\text{Weight loss (\%)} = (\text{Weight original} - \text{Weight measurement}) / \text{Weight original} \times 100\% \quad (2)$$

### 2.9. Total Phenolic Content

The total phenolic content (TPC) was determined using a modified version of the Folin–Ciocalteu method described by previous research [25]. Frozen pulp (1.0 g) was mixed with 5.0 mL of 70% ($v/v$) ethanol in a pre-cooled mortar and homogenized at a low temperature. The mixture was then centrifuged at $16,000 \times g$ for 10 min at 4 °C, and the resulting supernatant was collected as the extract. To 0.15 mL of the above extract, 1.5 mL of Folin–Ciocalteu reagent (diluted 10 times) was added, mixed and stood by for 5 min. Subsequently, 1.5 mL of 6% ($w/v$) sodium carbonate solution was added to the mixture, followed by incubation in a water bath at 75 °C for 10 min. The solution was then rapidly cooled in an ice bath for 30 s. The absorbance was measured at 725 nm (SpectraMax M2, Molecular Devices, San Jose, USA), using the extraction solvent as the blank. Gallic acid was used the standard substance, and the results were expressed as mg kg$^{-1}$ on a fresh weight basis.

### 2.10. Statistical Analysis

Principal component analysis (PCA) was applied to analyze the fruit quality parameters, which referred to a previous study [26]. The data were processed using Origin 2017 (OriginLab Corporation, Northampton, MA, USA) to calculate the eigenvalues, contribution, and factor scores (FAC) of the principal components. The Fernandez–Garcia definition was applied to explain the significance of each principal component in explaining the overall variance. This involved calculating the individual $F$-value (3) based on the FAC and eigenvalue, as well as the $F$-value for each observation (4). The average $F$-value was utilized as a ranking metric for comparing the different groups.

$$F_n = FAC_n \times \sqrt{\text{Eigenvalue}_n} \quad (3)$$

$$F\text{-value} = (F_1 \times Variance_1 + \cdots + F_n \times Variance_n) / \text{Cumulative} \quad (4)$$

All the data were organized and graphed using Excel (Microsoft Corp., Seattle, WA, USA), then analyzed using JMP statistical analysis software (version 16; SAS Institute, Cary, NC, USA). Analysis of variance (ANOVA) was used to evaluate the effect of the coating on guava quality, and Duncan's multiple range test with Holm correction was applied to determine significant differences ($p < 0.05$) among different groups at the same time point. At least three replications were conducted for all experiments to provide guava quality data.

## 3. Results and Discussion

### 3.1. Particle Size, Zeta Potential, and Polydispersity Index of Coating Solution

The application of edible coatings depends on different scenarios, such as immersion, spray, and artificial methods [27]. Therefore, the efficiency of the coating is related to the properties of its forming solution. Particle size is an important indicator of the uniformity and stability of the solution. The particle size of the chitosan solution was the smallest (Table 1), and the incorporation of carvacrol significantly increased the particle size of the solution ($p < 0.05$). This may be due to the aggregation of essential oil droplets in the solution.

Similarly, the addition of cinnamon essential oil increased the particle size of the chitosan film solution [16]. Emulsions can be classified into coarse emulsions (200 nm–200 μm) and nanoemulsions (0–200 nm) based on their particle size. Nanoemulsions have smaller droplet diameters, allowing for better distribution within the polysaccharide matrix and preserving the original membrane matrix structure, thereby minimizing adverse effects on membrane performance. Furthermore, nanoemulsions exhibit greater stability compared to macroemulsions and possess a higher surface area ratio, resulting in a slower release rate of essential oils and improved bioavailability [27]. The particle sizes of the prepared emulsions were all below 200 nm, which are consistent with previous reports [28,29] and can be applied as edible nano-coatings.

**Table 1.** Particle size, zeta potential, and polydispersity index of coating solutions *.

|  | Particle Size (nm) | Zeta Potential (mV) | Polydispersity Index |
| --- | --- | --- | --- |
| Chitosan | 127.3 $\pm$ 2.62 c | 55.46 $\pm$ 1.736 a | 0.22 $\pm$ 0.04 a |
| Chitosan + 0.1% Carvacrol | 144.3 $\pm$ 5.62 b | 52.33 $\pm$ 2.039 ab | 0.29 $\pm$ 0.06 a |
| Chitosan + 0.2% Carvacrol | 186.4 $\pm$ 8.80 a | 49.50 $\pm$ 3.224 b | 0.31 $\pm$ 0.07 a |

* The value is expressed as mean $\pm$ standard deviation ($n = 5$), the experiment was carried out three independent times. The different letters indicate significant differences (ANOVA, $p < 0.05$) among different groups in each parameter according to Duncan's multiple comparisons test.

The zeta potentials of all the samples were positive and over 40 mV (Table 1), indicating that the formed nanoemulsion has high stability. The polydispersity index was between 0.22 and 0.31, indicating that the essential oil was distributed uniformly in the emulsion, the applied emulsification in the experiment was effective, and the prepared nanoemulsion could be applied as a coating.

Zeta potential is an important indicator for evaluating the stability of the emulsion system. Positive values above +30 mV and negative values below $-30$ mV indicate that the particles in the system are stable. Chitosan amino groups are protonated at low pH values, resulting in their zeta potential being positive [29,30]. With the increasing carvacrol concentration, the zeta potential was observed to decrease significantly ($p < 0.05$), which may be due to the interaction of the carvacrol droplets with the free amino groups on the chitosan molecules, resulting in a slight decrease in stability [30]. PDI is an important parameter for characterizing the uniformity of the particle size distribution in emulsion systems. Dynamic light scattering (DLS) can be used to determine the size distribution profile of small particles in suspension or polymers in solution [31]. In this sense, the dispersity values are in the range from 0 to 1, with values between 0.1 and 0.25 indicating a narrow particle size distribution, while values above 0.5 indicate a wide particle size distribution. The PDIs of obtained nanoemulsions ranged from 0.22 to 0.31, which is within the range of the previous study [32], indicating a good distribution uniformity.

### 3.2. Fruit Appearance and Color Change

The fruit epidermis gradually turned yellow during storage, which is mainly due to the postharvest physiological process of the fruit, producing ethylene to promote the ripening of the fruit and cause the chlorophyll to decompose [33]. The control fruits were turning yellow on day 4 and exhibited wrinkling on day 8 (Figure 2A). The chitosan coating delayed yellowing, and it was slower than the control and no significant wrinkling occurred. The chitosan coating incorporating carvacrol significantly delayed fruit yellowing and reduced weight loss, maintaining the fruit appearance.

The biopolymer materials can form a selectively permeable film on the fruit surface, regulating the permeability of $O_2$ and enhancing the antioxidant system by respiratory metabolism regulation. The essential oil itself is hydrophobic, and the addition of the essential oil to the chitosan coating enhances the water barrier performance of the film and reduces the water loss of the fruit due to respiration [18]. In addition, the strong antioxidant property of carvacrol contributes to improving fruit quality [34]. These results

indicated that the carvacrol-incorporated coatings delayed senescence and maintained the fruit appearance. This result is consistent with the previous study, where the application of essential oils enhanced the barrier properties of chitosan coating, significantly decreasing the perishing process of postharvest tomatoes and strawberries [35].

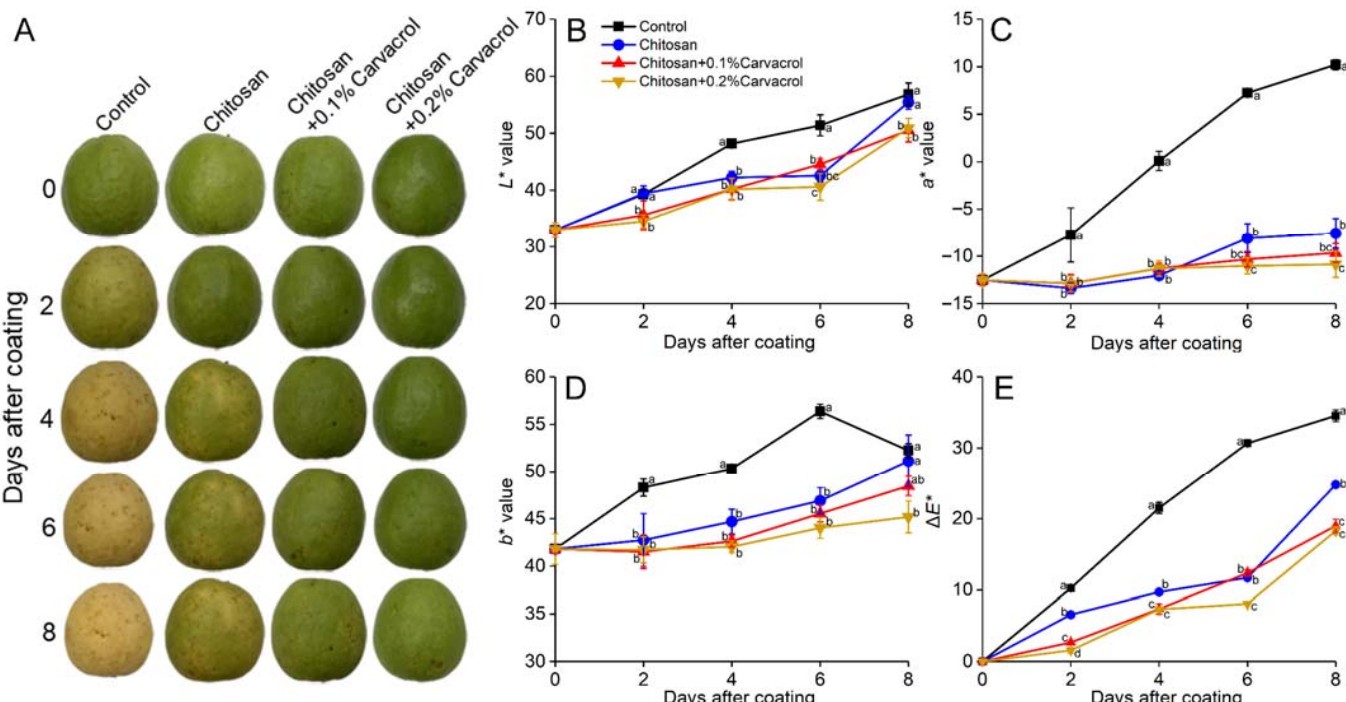

**Figure 2.** Effects of chitosan–carvacrol coating on appearance and peel color of postharvest guava fruit. The fruit appearance (**A**), peel $L^*$ (**B**), $a^*$ (**C**), $b^*$ (**D**), and $\Delta E^*$ (**E**) in response to chitosan–carvacrol coating. Each value is the mean of three replicates. The vertical bars represent the standard deviation of the means. The different letters indicate significant differences (ANOVA, $p < 0.05$) according to Duncan's multiple comparisons test with Holm correction.

The $a^*$ and $b^*$ of all groups increased continuously, showing the loss of green and the increasing yellow and red of the peel (Figure 2C,D). The chitosan coating delayed yellowing, and chitosan incorporating carvacrol further delayed this trend. After storage for 8 days, the $a^*$ of the control was the highest, which was 2.35-fold higher than the chitosan group; the $a^*$ of the chitosan + 0.1 and 0.2% carvacrol groups were 27.64% and 43.78% ($p < 0.05$) lower than chitosan alone, respectively. The $b^*$ of the control was also the highest during storage; at day 6, it was 20.24%, 23.81, and 28.08% ($p < 0.05$) higher than those of the chitosan and chitosan + 0.1 and 0.2% carvacrol groups, respectively. $\Delta E$ represents the total color difference compared with the initial state, and the $\Delta E$ of the control group was higher than the other three groups during storage, indicating that the color variation was greater, and the chitosan combined with carvacrol coating inhibited this significant change (Figure 2E). After 8 days of storage, the $\Delta E$ of the chitosan + 0.2% carvacrol group was the lowest, being 47.08% ($p < 0.05$), 26.22% ($p < 0.05$), and 3.63% lower than that of the control, chitosan, and chitosan + 0.1% carvacrol groups, respectively. These results indicated that the chitosan–carvacrol film could delay the color deterioration of the fruit epidermis, which was consistent with the conclusions of other polysaccharide–essential oil coating films [33,36].

In addition to controlling the fruit atmosphere, a small amount of chitosan entering the fruit will regulate the physiological metabolism and inhibit ripening and senescence [19]. Previous studies also reported that chitosan–essential oil coatings may regulate the activity of enzymes related to browning [37].

### 3.3. Fruit Firmness, Soluble Solid Content, Titratable Acidity, and Weight Loss

The firmness of guava decreased rapidly during storage, but the chitosan coating delayed this decrease. The control group showed the lowest firmness compared to that of the other three groups during the storage (Figure 3A). Both chitosan incorporated with 0.1% and 0.2% carvacrol coatings suppressed the decreasing firmness, which was 10.34% and 29.82% ($p < 0.05$) higher than that of chitosan alone after 8-day storage. The total soluble solid content (SSC) of the fruits increased slightly and then decreased during storage (Figure 3B). The SSC of the control fruit reached its peak on day 2 and then gradually decreased. The chitosan coating delayed the SSC peak until day 4, and it was 43.13% ($p < 0.05$) higher than the control at the end of storage. The consolidation of carvacrol further delayed the SSC decrease, its peak value was on day 6, and its content decreased slowly. Titratable acidity decreased constantly during storage, and the control group began to decrease rapidly from the second day, while all the coating groups remained at higher levels (Figure 3C). Chitosan + 0.2% carvacrol maintained the highest titratable acidity after 8 days, while the control was the lowest. The weight loss of all the groups increased constantly (Figure 3D). By establishing linear models of weight loss, the chitosan coating could slow down the weight loss of fruit from 2.46% per day in the control to 2.14%. The cooperation of carvacrol further inhibited weight loss, and the daily weight loss rates of chitosan + 0.1% and 0.2% carvacrol were 1.82% and 1.65%, respectively.

Firmness is one of the most important indicators of fruit quality, and its softening during postharvest storage is mainly due to the change in cell wall composition under the catalysis of various enzymes (such as polygalacturonase and pectin methylesterase), which decompose the middle lamella between cells. Oxygen is necessary for these enzymes, and the barrier characteristics of the coating may inhibit their activities [12]. The chitosan–carvacrol coatings delayed fruit ripening through respiratory regulation and suppressed the consumption of water, organic acids, and other substances. Previous studies have shown that chitosan coatings effectively reduced the respiration rate of fruit, maintaining SSC, titratable acidity, and weight loss [3,19]. Other coating materials also prolonged the shelf life of postharvest guavas. Gum arabic and *Aloe vera* gel extended guava shelf life, slowed weight loss, and resulted in higher titratable acidity [38], which is consistent with our results. The combination of a modified chitosan coating containing carvacrol nanoemulsions and pulsed light exhibited a high preservation efficiency on cucumber slices by producing positive effects on decontamination [39]. Further studies need to address the physiological changes in the fruit, including respiration rate, ethylene release rate, etc., to provide more comprehensive information on how the coating affects fruit physiology change. Since both chitosan and carvacrol exhibit antimicrobial properties, it is also important to study how the microbial parameters change in fruit surfaces/wounds, which provide decay and microbial-related information.

### 3.4. Total Phenolic Content

Polyphenols are important bioactive substances that contribute to the antioxidant activity of guava. During fruit ripening and senescence, polyphenols will gradually decrease, reducing the nutritional value of the fruit. Chitosan coatings incorporating carvacrol delayed the decline in total phenolic content (Figure 3E). On the last day of storage, the total phenolic content of the chitosan + 0.2% carvacrol was 17.14% ($p < 0.05$) higher than that of the control.

The decline in total phenolic content in guava is related to the higher respiration rate. Previous studies have shown that controlled atmosphere storage can slow down the ripening process and maintain the phenolic content [40]. The chitosan coatings prepared in this study may delay polyphenol decomposition by atmosphere regulation. Similarly, higher contents of polyphenols and flavonoids were observed in other edible coatings, maintaining the antioxidant capacity and nutritional value of the coated guavas [41]. Polyphenols have been shown to be directly associated with antimicrobial activity. They also serve as signal molecules involved in defense responses against stress, affecting

postharvest disease incidence [42]. Further study will determine whether the combination of chitosan and carvacrol in coatings affects polyphenol content by regulating antioxidant and antimicrobial activities, barrier properties, enzyme activities, pH alterations, and other potential pathways.

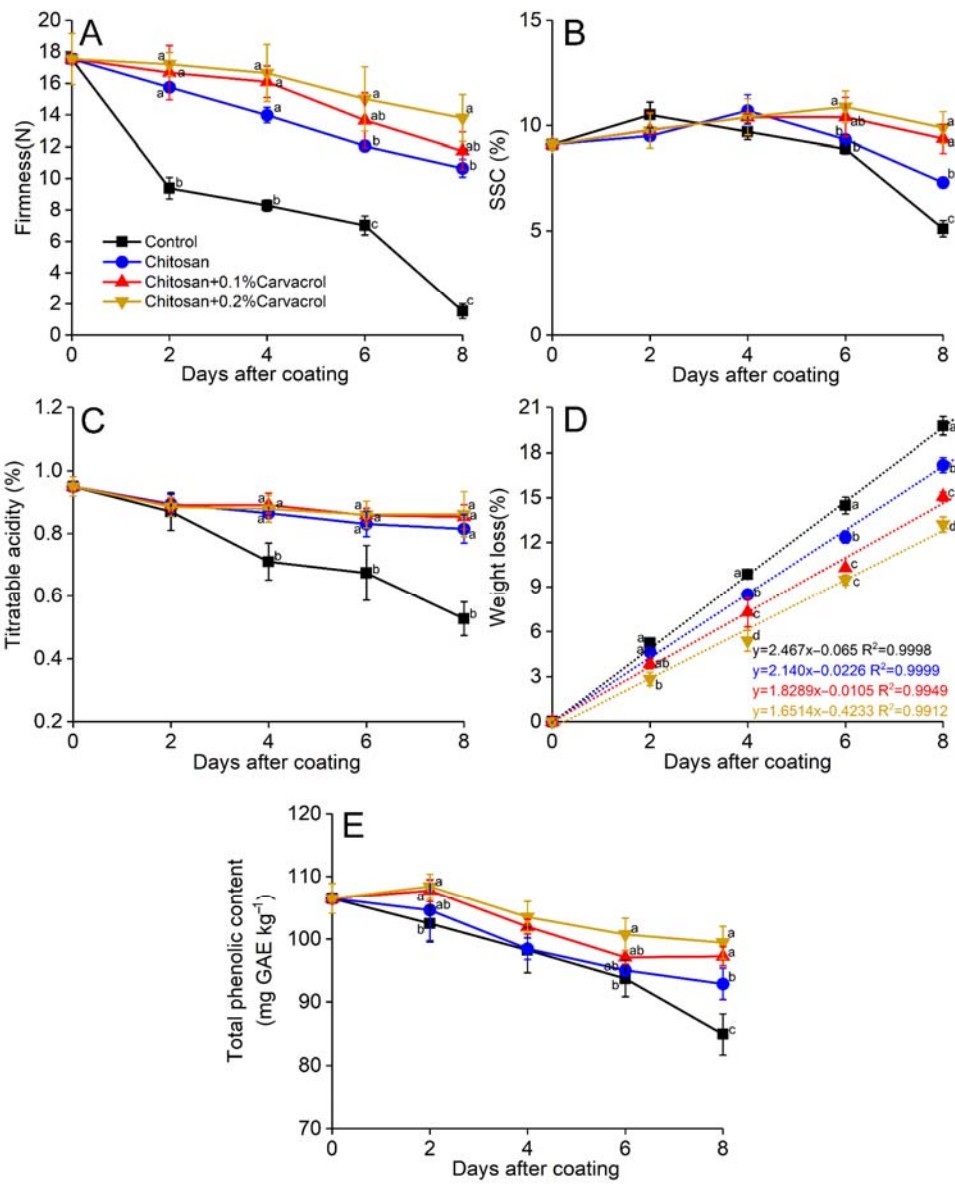

**Figure 3.** Effects of chitosan–carvacrol coating on quality parameters of postharvest guava fruit. Firmness (**A**), soluble solid content (SSC); (**B**), titratable acidity (**C**), weight loss (**D**), and total phenolic content (**E**). Each value is the mean of three replicates. The vertical bars represent the standard deviation of the means. The different letters indicate significant differences (ANOVA, $p < 0.05$) according to Duncan's multiple comparisons test with Holm correction.

### 3.5. Correlation and Principal Component Analysis of Fruit Quality Attribute Response to Chitosan–Carvacrol Coatings

Based on the data presented in Figure 4 and Table S1 in Supplementary Materials, it can be concluded that firmness, SSC, and TPC exhibited a negative correlation with weight loss and the color indexes. This suggests that with the fruit ripening, there was a consistent decrease in firmness, SSC, and TPC accompanied by an increase in weight loss and color parameters. The correlation between titratable acid and the other indicators appeared to be

relatively weaker; this may be due to cultivar differences. The initial titratable acid content was higher and remained higher during storage.

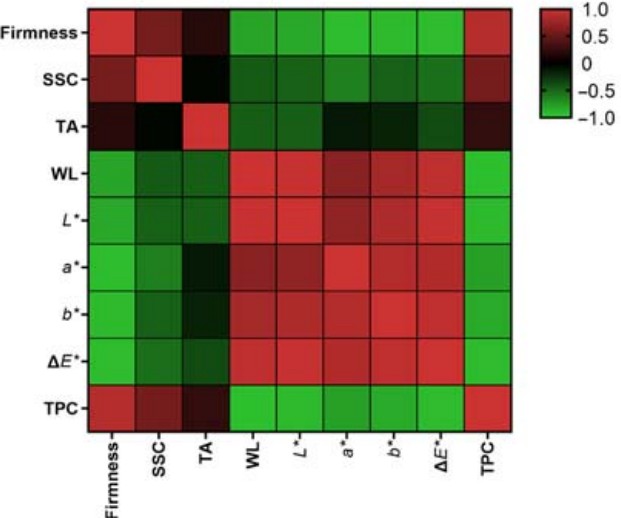

**Figure 4.** Pearson correlation analysis among the measured parameters. SSC: total soluble solid content; TA: titratable acidity; WL: weight loss; TPC: total phenolic content.

Principal component analysis (PCA) is a widely recognized method for reducing data dimensionality, which transforms a large amount of data into fewer dimensions while retaining the characteristics of the original data. All the measured indicators in this study were analyzed, and the cumulative contribution of the first five principal components obtained reached 99.0% (Table 2), which can explain the correlation characteristics of the data. Principal component 1 (PC1) explained 82.5% of the total variance and was primarily influenced by firmness, TA, and TPC. They clustered together in the score plot (Figure 5B), indicating close associations. PC2 explained 8.5% of the total variance of the original variables and was determined by the weight loss and color parameters, indicating a strong relationship between color variation and fruit weight loss. The relationship between these indicators was also observed in the PCA analysis of kiwi [43] and pear [44]. In the PC1 direction, there was a good distinction between the different storage times for each group (Figure 5A). At the beginning of storage, observation points were located in the negative half-axis of PC1, indicating good quality attributes. However, with prolonged storage, observations gradually shifted toward the positive half-axis of PC1. Notably, the last two data points of the control group were on the rightmost side of the PC1 axis, indicating the most significant quality deterioration. During the ripening, the SSC initially increased to a peak and then gradually decreased. The Cvc0.2_8 was located at the top of the positive axis of PC2, while the Con_8 treatment was at the negative half-axis of PC2, suggesting the SSC in the control group significantly decreased while the coating treatments maintained a higher level of SSC at the same time point, which was consistent with the actual measurement result.

**Table 2.** Eigenvalues and contribution rate of the principal components.

| Principle Component | Eigenvalue | Percentage of Variance (%) | Cumulative (%) |
|---|---|---|---|
| 1 | 7.423 | 82.479 | 82.479 |
| 2 | 0.762 | 8.467 | 90.945 |
| 3 | 0.471 | 5.238 | 96.183 |
| 4 | 0.160 | 1.780 | 97.963 |
| 5 | 0.097 | 1.083 | 99.046 |

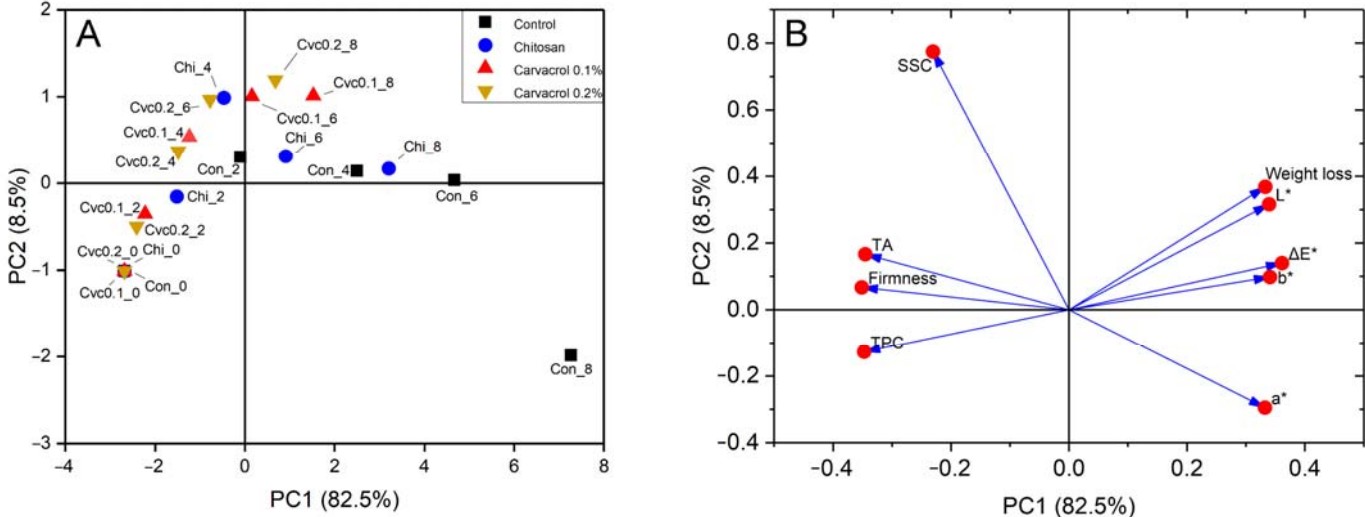

**Figure 5.** Principal component analysis (PCA) results of measured parameters. Score plot (**A**), loading plot (**B**). Con: control; Chi: chitosan; Cvc: carvacrol; SSC: total soluble solid content; TA: titratable acidity; TPC: total phenolic content. The number following the underscore indicates days after coating.

During the storage from day 2 to 8, the enclosed area of the control exhibited the greatest distance from the initial storage point, indicating a significant deterioration in fruit quality (Figure S1). The chitosan treatment shortened its enclosed area's distance on day 0, indicating an improvement in storage quality compared to the control. However, there was still some overlap between the chitosan treatment and the control group, suggesting that the improvement was not as pronounced. In contrast, the chitosan coating containing carvacrol was the closest to the initial point, and there was no overlap with the ensemble area of the control, indicating that the incorporation of carvacrol significantly improved fruit quality. This indicated that the chitosan coating containing carvacrol significantly enhanced the quality of the fruit. A similar result was also observed in the PCA conducted on the quality of litchi [45]. Considering the principal component score and loadings plots, chitosan coatings delayed fruit senescence, while the incorporation of chitosan and carvacrol coatings maintained better fruit quality compared to other treatment groups.

By evaluating the *F*-values of the component scores derived from PCA, *F*-values increased for each group with extended storage time, indicating quality deterioration compared to the initial state (Table 3). Based on the mean *F*-value average for each group, the four groups could be ranked in descending order: control > chitosan > chitosan + 0.1% carvacrol > chitosan + 0.2% carvacrol. This suggested that the control group experienced the most significant quality deterioration compared to the initial state of the fruit, while the chitosan + 0.2% carvacrol group exhibited the slightest change. This demonstrated that chitosan coatings incorporating carvacrol delayed the ripening and maintained the postharvest quality of guava fruit.

Further study is needed for the characterization and optimization of the coating, and more fruit physiological parameters need to be measured to provide more comprehensive insights into the coating's effectiveness. Also, the combination of chitosan and essential oil, multiple-layer coatings, and other coating materials should be further studied.

**Table 3.** Principal component score.

| Group | Storage Time (d) | Name of Observations | FAC1 | FAC2 | FAC3 | FAC4 | FAC5 | F1 | F2 | F3 | F4 | F5 | F | F Average |
|---|---|---|---|---|---|---|---|---|---|---|---|---|---|---|
| Control | 0 | Con_0 | −2.680 | −1.010 | −0.086 | 0.234 | −0.104 | −7.299 | −0.880 | −0.059 | 0.094 | −0.031 | −6.153 | |
| | 2 | Con_2 | −0.094 | 0.304 | 1.245 | 0.485 | −0.483 | −0.255 | 0.265 | 0.854 | 0.194 | −0.145 | −0.185 | |
| | 4 | Con_4 | 2.497 | 0.139 | 1.244 | −0.065 | 0.338 | 6.802 | 0.121 | 0.853 | −0.026 | 0.101 | 5.678 | 5.253 |
| | 6 | Con_6 | 4.668 | 0.036 | 1.396 | 0.604 | 0.234 | 12.716 | 0.032 | 0.957 | 0.242 | 0.070 | 10.596 | |
| | 8 | Con_8 | 7.265 | −1.990 | −0.449 | −0.763 | −0.087 | 19.790 | −1.735 | −0.308 | −0.305 | −0.026 | 16.329 | |
| Chitosan | 0 | Chi_0 | −2.680 | −1.010 | −0.086 | 0.234 | −0.104 | −7.299 | −0.880 | −0.059 | 0.094 | −0.031 | −6.153 | |
| | 2 | Chi_2 | −1.513 | −0.156 | −0.274 | 0.007 | 0.203 | −4.123 | −0.136 | −0.188 | 0.003 | 0.061 | −3.445 | |
| | 4 | Chi_4 | −0.467 | 0.989 | 0.062 | −0.325 | −0.358 | −1.272 | 0.862 | 0.042 | −0.130 | −0.107 | −0.985 | −0.243 |
| | 6 | Chi_6 | 0.912 | 0.313 | −0.258 | −0.227 | −0.699 | 2.484 | 0.272 | −0.177 | −0.091 | −0.210 | 2.090 | |
| | 8 | Chi_8 | 3.206 | 0.169 | −1.556 | 0.817 | 0.014 | 8.734 | 0.147 | −1.066 | 0.327 | 0.004 | 7.281 | |
| Chitosan + 0.1% Carvacrol | 0 | Cvc0.1_0 | −2.680 | −1.010 | −0.086 | 0.234 | −0.104 | −7.299 | −0.880 | −0.059 | 0.094 | −0.031 | −6.153 | |
| | 2 | Cvc0.1_2 | −2.222 | −0.353 | 0.071 | −0.214 | 0.355 | −6.053 | −0.308 | 0.049 | −0.086 | 0.106 | −5.066 | |
| | 4 | Cvc0.1_4 | −1.235 | 0.538 | −0.063 | −0.361 | 0.042 | −3.364 | 0.469 | −0.043 | −0.144 | 0.013 | −2.761 | −2.002 |
| | 6 | Cvc0.1_6 | 0.159 | 1.007 | −0.107 | −0.241 | −0.294 | 0.433 | 0.877 | −0.074 | −0.097 | −0.088 | 0.435 | |
| | 8 | Cvc0.1_8 | 1.526 | 1.014 | −0.679 | 0.403 | 0.052 | 4.156 | 0.884 | −0.465 | 0.161 | 0.016 | 3.534 | |
| Chitosan + 0.2% Carvacrol | 0 | Cvc0.2_0 | −2.680 | −1.010 | −0.086 | 0.234 | −0.104 | −7.299 | −0.880 | −0.059 | 0.094 | −0.031 | −6.153 | |
| | 2 | Cvc0.2_2 | −2.410 | −0.500 | 0.181 | −0.173 | 0.377 | −6.564 | −0.436 | 0.124 | −0.069 | 0.113 | −5.502 | |
| | 4 | Cvc0.2_4 | −1.481 | 0.372 | 0.067 | −0.408 | 0.310 | −4.034 | 0.324 | 0.046 | −0.163 | 0.093 | −3.332 | −3.008 |
| | 6 | Cvc0.2_6 | −0.775 | 0.967 | 0.206 | −0.470 | −0.111 | −2.111 | 0.843 | 0.141 | −0.188 | −0.033 | −1.686 | |
| | 8 | Cvc0.2_8 | 0.681 | 1.191 | −0.741 | −0.004 | 0.523 | 1.856 | 1.039 | −0.508 | −0.002 | 0.157 | 1.632 | |

## 4. Conclusions

The result illustrated that a 2% (*w/v*) chitosan coating containing 0.1–0.2% carvacrol (*v/v*) maintained the postharvest quality of guava compared to the chitosan alone, with higher firmness, soluble solid content, titratable acid, and total phenolic content, with lower weight loss and peel yellowing. The results were verified by correlation and PCA analysis. Chitosan coatings containing carvacrol can be applied as an effective strategy to improve postharvest guava and other fruit quality.

**Supplementary Materials:** The following supporting information can be downloaded at https://www.mdpi.com/article/10.3390/horticulturae10010080/s1, Table S1: The R-value of Pearson correlation analysis among the measured parameters; Figure S1: Principal component score plots containing enclosed areas of each group.

**Author Contributions:** Conceptualization, X.S.; methodology, X.S. and C.S.; software, C.S.; validation, C.S. and X.S.; formal analysis, C.S.; investigation, C.S., B.K.-L. and X.S.; resources, X.S.; data curation, X.S.; writing—original draft preparation, C.S.; writing—review and editing, C.S. and X.S.; visualization, C.S.; supervision, X.S.; project administration, X.S.; funding acquisition, X.S. All authors have read and agreed to the published version of the manuscript.

**Funding:** This research received no external funding.

**Data Availability Statement:** The original contributions presented in the study are included in the article and supplementary materials, further inquiries can be directed to the corresponding author.

**Acknowledgments:** This research was supported in part by an appointment to the Agricultural Research Service (ARS) Research Participation Program administered by the Oak Ridge Institute for Science and Education (ORISE) through an interagency agreement between the U.S. Department of Energy (DOE) and the U.S. Department of Agriculture (USDA). ORISE is managed by ORAU under DOE contract number DE-SC0014664. All opinions expressed in this paper are the author's and do not necessarily reflect the policies and views of the USDA, DOE, or ORAU/ORISE. The USDA is an equal opportunity provider and employer.

**Conflicts of Interest:** The authors declare no conflicts of interest.

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
