# Peer review of "Chitosan Coating Incorporated with Carvacrol Improves Postharvest Guava (Psidium guajava) Quality"

_horticulturae, doi:10.3390/horticulturae10010080_

Round 1
Reviewer 1 Report
Comments and Suggestions for Authors
The manuscript investigated the effect of chitosan complexed with carvacrol on the storage quality of Guava. The results of the study showed that chitosan nano-emulsion containing 0.2% carvacrol could effectively maintain the storage quality. The results may provide new options for postharvest preservation to minimize Guava loss. Here are just a few questions that need to be answered.
1. What is the temperature at which chitosan is dissolved, what is the temperature at which the coating is applied, and how does manual coating application ensure consistent coating thickness of fruit?
2. What is the main purpose of adding carvacrol? Preservation of freshness or bacteriostasis? Are there any indicators related to disease incidence measured?
Author Response
Comment 1: The manuscript investigated the effect of chitosan complexed with carvacrol on the storage quality of Guava. The results of the study showed that chitosan nano-emulsion containing 0.2% carvacrol could effectively maintain the storage quality. The results may provide new options for postharvest preservation to minimize Guava loss. Here are just a few questions that need to be answered.
Response 1: Thank you very much for your positive comments. We have revised the manuscript according to your suggestions. Our point-to-point responses are as follows.
Comment 2: What is the temperature at which chitosan is dissolved, what is the temperature at which the coating is applied, and how does manual coating application ensure consistent coating thickness of fruit?
Response 2: Chitosan was dissolved at 23°C in distilled water, the subsequent homogenization process may generate heat, we put the homogenized coating solution at ambient temperature till reach 23°C to make sure the fruit peel would not be injured by the heat. The fruit for the experiment were similar in weight and size (130 ± 15 g), all the fruit were coated with 1.0 mL of the coating solution to ensure consistent coating on each fruit. We added the corresponding details in the method section of the manuscript.
Comment 3: What is the main purpose of adding carvacrol? Preservation of freshness or bacteriostasis? Are there any indicators related to disease incidence measured?
Response 3: Carvacrol is a plant essential oil with antioxidant and antimicrobial activity. It has been proven on many fruit to delay postharvest ripening and senescence and inhibit microbial growth. At the same time, as a fat-soluble component, it can increase the hydrophobicity of chitosan coating and reduce fruit weight loss due to respiration. So, the purpose of adding carvacrol is to preserve freshness. We mainly measured indicators related to postharvest quality in this study, but our further study measured the impact of the coating on total microbial community on the fruit surface. We have added a discussion illustrating the need for further determination of disease related indicators in the corresponding section of the text.
Reviewer 2 Report
Comments and Suggestions for Authors
I have reviewed the manuscript Chitosan Coating Incorporated with Carvacrol Improves Postharvest Guava (Psidium guavaja) Quality and I have some observations.
I suggest to the authors that include more physiological analysis such as respiration rate, ethylene production, etc.
If they need to demonstrate that the phenolic compounds are conserved using the coating, I suggest that they include a phenolic profile, it will give more certainy.
The results only are descriptive the authors do not made a good discussion of them.
Author Response
Comment 1: I have reviewed the manuscript Chitosan Coating Incorporated with Carvacrol Improves Postharvest Guava (Psidium guavaja) Quality and I have some observations.
Response 1: Thanks for your review, we revised the manuscript according to your suggestions. Our point-to-point responses are as follows.
Comment 2: I suggest to the authors that include more physiological analysis such as respiration rate, ethylene production, etc.
Response 2: Thank you for your valuable suggestion. In this study, we prepared the coating and tested its impact on the quality attributes of postharvest guavas. We further employed a PCA model to evaluate the overall effects of the coating on fruit quality. Therefore, the overall quality is the focus in this study. Our further study addressed the physiological measurements of fruit, focusing on physiological aspects including respiration rate, ethylene release rate, ROS content, and relative conductivity. Acknowledging the importance of physiological parameters, we have included additional discussion on how the chitosan coating influences these physiological changes in postharvest fruit.
Comment 3: If they need to demonstrate that the phenolic compounds are conserved using the coating, I suggest that they include a phenolic profile, it will give more certainy.
Response 3: We measured the total phenolic content in this research, which affects the overall quality of the fruit. We further measured the phenolic profile change in up-coming study. We included additional discussion on phenolic profile change and its potential role in fruit physiological change.
Comment 4: The results only are descriptive the authors do not made a good discussion of them.
Response 4: Thank you for your suggestion, we improved the discussion in each section, please check in the text.
Reviewer 3 Report
Comments and Suggestions for Authors
The study focuses on the use of chitosan-essential oil nano-emulsions as an edible coating to enhance postharvest quality attributes of guava fruit. This topic is relevant because guava is an economically important crop grown widely in tropical and subtropical regions, and is prone to rapid ripening and senescence, leading to short shelf life and quality deterioration. The authors successfully prepared a nano-emulsoid solution containing chitosan and carvacrol, and confirmed the stability of the emulsion through ζ potential and polydispersity index measurements.
Overall, the abstract provides a clear overview of the study's objectives, methods, and findings. However, there are a few suggestions for improvement:
1.The authors mention that chitosan entering the fruit may regulate physiological metabolism and inhibit ripening and senescence, but it is not clear how much chitosan actually enters the fruit. Including quantitative data on chitosan uptake by the fruit would strengthen the study.
2.The use of multiple comparisons without correcting for multiple testing increases the risk of type I error. The authors should consider applying a correction method (such as Bonferroni or Holm) to control for multiple testing.
3.The authors mention that chitosan coating incorporated with carvacrol can delay the decline in total phenolic content. However, further clarification is required on how chitosan and carvacrol work synergistically to slow down the degradation of polyphenols.
4.Previous studies have demonstrated that controlled atmosphere storage can slow down the ripening process and maintain polyphenol content. Nevertheless, more information is needed regarding the effects of chitosan coatings and a comparison with previous research findings. The text highlights that other edible coatings exhibit higher levels of polyphenols and flavonoids, thereby preserving the antioxidant capacity and nutritional value of coated guavas. Consequently, a more detailed comparison and discussion of the effects of different coating materials is necessary.
5.The authors must be included new, relevant and more information about other chitosan maters. The following reference can be included in the Introduction part to improve the quality of manuscript, because it provides relevant information:
DOI: 10.1016/j.ijbiomac.2020.09.034
DOI: 10.1016/j.ijfoodmicro.2017.08.011
Author Response
Comment 1: The study focuses on the use of chitosan-essential oil nano-emulsions as an edible coating to enhance postharvest quality attributes of guava fruit. This topic is relevant because guava is an economically important crop grown widely in tropical and subtropical regions, and is prone to rapid ripening and senescence, leading to short shelf life and quality deterioration. The authors successfully prepared a nano-emulsoid solution containing chitosan and carvacrol, and confirmed the stability of the emulsion through ζ potential and polydispersity index measurements. Overall, the abstract provides a clear overview of the study's objectives, methods, and findings. However, there are a few suggestions for improvement:
Response 1: Thanks for your review, we revised the manuscript according to your suggestions. Our point-to-point responses are as follows.
Comment 2: The authors mention that chitosan entering the fruit may regulate physiological metabolism and inhibit ripening and senescence, but it is not clear how much chitosan actually enters the fruit. Including quantitative data on chitosan uptake by the fruit would strengthen the study.
Response 2: In this study, the molecular weight of chitosan is between 310,000-375,000 Daltons, it can form a uniform coating on the fruit surface. Previous studies have demonstrated that chitosan can act on wounds and be taken up by plants through leaves, wounds, and stomata, thereby inducing defense responses and regulating physiological metabolism (Mukarram et al., 2023). However, current research does not have a reliable method to determine exactly how much chitosan is taken up by the plant/fruit. In order to make sure that all fruit were treated with the same dose of chitosan, previous studies soaked/sprayed the fruit in the chitosan solution at the same time. In our case, the fruit of uniform size and shape were used in the experiment, and the same volume and concentration of the coating solution coated each fruit. Therefore, what we can do at present is to maintain the homogeneity of chitosan coating on the fruit surface, and currently, there is no method to determine its content accurately.
Reference:
Mukarram, M., Ali, J., Kurjak, D., Kačík, F., & Ďurkovič, J. (2023). Chitosan-induced biotic stress tolerance and crosstalk with phytohormones, antioxidants, and other signalling molecules. Frontiers in Plant Science, 14, 1217822. https://doi.org/10.3389/fpls.2023.1217822
Comment 3: The use of multiple comparisons without correcting for multiple testing increases the risk of type I error. The authors should consider applying a correction method (such as Bonferroni or Holm) to control for multiple testing.
Response 3: We re-analyzed the multiple comparisons with Holm correction. We revised in the Method section, and the figure captions.
Comment 4: The authors mention that chitosan coating incorporated with carvacrol can delay the decline in total phenolic content. However, further clarification is required on how chitosan and carvacrol work synergistically to slow down the degradation of polyphenols.
Response 4: Chitosan and carvacrol can act synergistically to delay the degradation of polyphenols in postharvest produces. This synergistic effect can be attributed to several mechanisms: (1) antioxidant properties: both chitosan and carvacrol possess antioxidant properties. Carvacrol is a phenolic compound known for its strong antioxidant activity, which can directly scavenge free radicals and thus protect polyphenols from oxidative degradation. Chitosan, although less potent as an antioxidant compared to carvacrol, can contribute to the overall antioxidant capacity, thereby providing a more stable environment for the preservation of polyphenols; (2) enhanced barrier properties: chitosan forms a semi-permeable coating on the surface of fruit, which can reduce the rate of respiration and transpiration. This barrier can help maintain a more stable internal environment, reducing oxidative stress and slowing down the metabolic processes that lead to the degradation of polyphenols; (3) antimicrobial activity: both chitosan and carvacrol exhibit antimicrobial properties. The presence of these compounds can inhibit the growth of microorganisms that may contribute to the degradation of polyphenols. Carvacrol is particularly effective against a wide range of pathogens, and its incorporation into chitosan coatings can enhance the antimicrobial efficacy of the coating; (4) synergy in coating matrix: chitosan and carvacrol can interact within the matrix to create a more effective barrier. Carvacrol can be encapsulated or dispersed within the chitosan matrix, which can help in the controlled release of carvacrol, thereby prolonging its antioxidant and antimicrobial effects; (5) regulation of enzyme activity: certain enzymes, such as polyphenol oxidase (PPO), are responsible for the degradation of polyphenols. Chitosan coatings can act as a barrier to oxygen, reducing the substrate availability for these enzymes, while carvacrol's antioxidant properties can further inhibit the enzymatic oxidation of polyphenols; (6) pH alteration: chitosan can alter the pH on the surface of the coated produce. A lower pH environment, which is more acidic, can be conducive to the stability of polyphenols. Carvacrol, being a phenolic compound, may also contribute to this effect. Our up-coming research measured the physiological pathway-related enzyme activity (Lunkov et al., 2018; Alnadari et al., 2022; Xu et al., 2023). We addressed corresponding discussion in the text.
Reference:
Lunkov, A. P., Ilyina, A. V., & Varlamov, V. P. (2018). Antioxidant, antimicrobial, and fungicidal properties of chitosan based films. Applied Biochemistry and Microbiology, 54, 449-458.
Alnadari, F., Bassey, A. P., Abdin, M., Salama, M. A., Nasiru, M. M., Dai, Z., ... & Zeng, X. (2022). Development of hybrid film based on carboxymethyl chitosan-gum arabic incorporated citric acid and polyphenols from Cinnamomum camphora seeds for active food packaging. Journal of Polymers and the Environment, 30(9), 3582-3597.
Xu, Z., Liu, G., Zheng, L., & Wu, J. (2023). A polyphenol-modified chitosan hybrid hydrogel with enhanced antimicrobial and antioxidant activities for rapid healing of diabetic wounds. Nano Research, 16(1), 905-916.
Comment 5: Previous studies have demonstrated that controlled atmosphere storage can slow down the ripening process and maintain polyphenol content. Nevertheless, more information is needed regarding the effects of chitosan coatings and a comparison with previous research findings. The text highlights that other edible coatings exhibit higher levels of polyphenols and flavonoids, thereby preserving the antioxidant capacity and nutritional value of coated guavas. Consequently, a more detailed comparison and discussion of the effects of different coating materials is necessary.
Response 5: Thank you for your suggestion. The purpose of this study is to prepare the chitosan-carvacrol coating, and to test its effects on postharvest guavas. Our further study also applying chitosan coating as a matrix to improve shelf-life of postharvest guavas. Regarding other coating materials, previous studies already have a wide range of studies. For instance, hydrocolloids are the most common group of biopolymers used in the production of edible materials. Cellulose derivatives, starch, alginate, pectin, chitosan, pullulan, and carrageenan are the most popular polysaccharides used in the production of edible films and coating, whereas among proteins the most popular are soybean proteins, wheat gluten, corn zein, sunflower proteins, gelatin, whey, casein, and keratin. Those materials are hydrophilic, different types of oils and fats are incorporated into the hydrocolloid matrix in order to enhance their water vapor barrier properties. The most popular are waxes, triglycerides, acetylated monoglycerides, free fatty acids, and vegetable oils. However, there are huge differences between different coating materials, which allows them to be adapted to different fruit. For example, water vapor transmission rate, oxygen transmission rate, affinity to the fruit surface, etc. will all affect its applicability. Extensive previous research has shown that chitosan is an ideal material for guava preservation and can significantly improve postharvest quality attributes, extend shelf life (Silva et al., 2018), and improve overall quality (Nair et al., 2018). Chitosan can also be used as a matrix to combine with different active compounds such as antimicrobials, antioxidant, color agents, flavors, and nutraceuticals are incorporated into film-forming solution in order to improve the quality, stability, and safety of packed foods. In addition, those ingredients may provide antibacterial, antifungal or antioxidant properties of edible material (Rojas-Graü et al., 2009; Salgado et al., 2015). So, in this study, we focus on the chitosan coatings only, the comparation with other coating material is difficult to control the variables for the whole study. We addressed some discussion regarding other coating materials on guava fruit.
Reference:
Falguera, V., Quintero, J. P., Jiménez, A., Muñoz, J. A., & Ibarz, A. (2011). Edible films and coatings: Structures, active functions and trends in their use. Trends in Food Science & Technology, 22(6), 292-303.
Galus, S., & Kadzińska, J. (2015). Food applications of emulsion-based edible films and coatings. Trends in Food Science & Technology, 45(2), 273-283.
Rojas-Graü, M. A., Soliva-Fortuny, R., & Martín-Belloso, O. (2009). Edible coatings to incorporate active ingredients to fresh-cut fruits: a review. Trends in food science & technology, 20(10), 438-447.
Nair, M., Saxena, A., & Kaur, C. (2018). Effect of chitosan and alginate based coatings enriched with pomegranate peel extract to extend the postharvest quality of guava (Psidium guajava L.). Food chemistry, 240, 245-252. https://doi.org/10.1016/j.foodchem.2017.07.122.
Salgado, P. R., Ortiz, C. M., Musso, Y. S., Di Giorgio, L., & Mauri, A. N. (2015). Edible films and coatings containing bioactives. Current Opinion in Food Science, 5, 86-92.
Silva, W., Silva, G., Santana, D., Salvador, A., Medeiros, D., Belghith, I., Silva, N., Cordeiro, M., & Misobutsi, G. (2018). Chitosan delays ripening and ROS production in guava (Psidium guajava L.) fruit. Food chemistry, 242, 232-238. https://doi.org/10.1016/j.foodchem.2017.09.052.
Comment 6: The authors must be included new, relevant and more information about other chitosan maters. The following reference can be included in the Introduction part to improve the quality of manuscript, because it provides relevant information: DOI: 10.1016/j.ijbiomac.2020.09.034, DOI: 10.1016/j.ijfoodmicro.2017.08.011
Response 6: We cited the recommend articles in the introduction.
Reviewer 4 Report
Comments and Suggestions for Authors
The manuscript presented a study on the effectiveness of chitosan-carvacrol coating in enhancing the postharvest quality of guava. The study examines various parameters such as fruit firmness, soluble solid content, titratable acidity, weight loss, and total phenolic content. Here are some detailed points:
- At line 188, the term "polydispersity index" is mentioned with a value under one. This could cause confusion, particularly for readers with a polymer background, where PDI is defined as the ratio of weight average molecular weight (Mw) to number average molecular weight (Mn). A clearer, more specific definition or context for PDI in this study should be provided to avoid ambiguity.
- The oxygen barrier properties are crucial in maintaining fruit freshness. Therefore, it is recommended that Oxygen Transmission Rate (OTR), Water Vapor Transmission Rate (WVTR), and antimicrobial properties of the coating be evaluated. Additionally, measuring the coating's thickness is essential, as it significantly influences these properties. Since measuring the thickness directly on fruits might be challenging, using a polymer substrate as a proxy could facilitate these tests, providing more comprehensive insights into the coating's effectiveness.
Author Response
Comment 1: The manuscript presented a study on the effectiveness of chitosan-carvacrol coating in enhancing the postharvest quality of guava. The study examines various parameters such as fruit firmness, soluble solid content, titratable acidity, weight loss, and total phenolic content. Here are some detailed points:
Response 1: Thank you very much for your comments. We have revised the manuscript according to your suggestions. Our point-to-point responses are as follows.
Comment 2: At line 188, the term "polydispersity index" is mentioned with a value under one. This could cause confusion, particularly for readers with a polymer background, where PDI is defined as the ratio of weight average molecular weight (Mw) to number average molecular weight (Mn). A clearer, more specific definition or context for PDI in this study should be provided to avoid ambiguity.
Response 2: The polydispersity index (PDI) is a measure of the heterogeneity of a sample based on molecular size. Polydispersity can occur due to size distribution in a sample or agglomeration or aggregation of the sample during isolation or analysis (Mudalige et al., 2019). The PDI in this study was obtained from the Zetasizer analyzer (Zetasizer Ultrablue, Malvern Instruments Ltd, Worcestershire, UK), which equipped with a dynamic light scattering (DLS). In this sense, the dispersity values are in the range from 0 to 1. The international standards organization (ISO) determined PDI value < 0.05 are more common to monodisperse samples, and value > 0.7 means a broad size (e.g., polydisperse) distribution of particles (ISO standards ISO 22,412:2017 and ISO 22,412:2017). To avoid ambiguity, we addressed more explanation regarding this item in the method section.
Reference:
Mudalige, T., Qu, H., Van Haute, D., Ansar, S. M., Paredes, A., & Ingle, T. (2019). Characterization of nanomaterials: Tools and challenges. Nanomaterials for food applications, 313-353.
Comment 3: The oxygen barrier properties are crucial in maintaining fruit freshness. Therefore, it is recommended that Oxygen Transmission Rate (OTR), Water Vapor Transmission Rate (WVTR), and antimicrobial properties of the coating be evaluated. Additionally, measuring the coating's thickness is essential, as it significantly influences these properties. Since measuring the thickness directly on fruits might be challenging, using a polymer substrate as a proxy could facilitate these tests, providing more comprehensive insights into the coating's effectiveness.
Response 3: This research focuses on the preparation of the coating and analyzing its effects on fruit edible quality using a PCA model. We have another work focusing on the characterization and the optimization of the coating, as you mentioned, we used 30% (w/w) glycerol as a plasticizer, and we further measured the oxygen transmission rate, water vapor transmission rate, antimicrobial properties against postharvest pathogens, and other physical parameters such as the color and thickness of the coating. These data are in our upcoming study, so we present more data on the fruit quality. We addressed the corresponding discussion in the manuscript to provide more comprehensive insights and further work.
Round 2
Reviewer 1 Report
Comments and Suggestions for Authors
The manuscript can be accepted in present form.
Comments on the Quality of English Languageno
Reviewer 2 Report
Comments and Suggestions for Authors
The authors must include the effect of coating on more specific physiological analysis and profile phenolic compounds, if not the quality of manuscript is very low. The PCA analysis is only one more tool for the statistical analysis of data but it no deepen about the physiological and biochemical changes from fruit when it is coated.
There is many references incluiding changes in basic analysis (pH, acidity, solid solubles, firmness, color, etc,) and total phenols when the fruits are coated and the behavior is very similar your results. Therefore, I consider mandatory that the authors must include more specific analysis
Author Response
Thank you for your insightful comments and suggestions regarding our manuscript. We greatly appreciate the time and effort you have invested in reviewing our work and providing constructive feedback. We understand your concern regarding the incorporation of more specific physiological analyses and profiling of phenolic compounds in our study. Indeed, such analyses can provide a deeper understanding of the biochemical changes in fruits when coated. However, we would like to clarify the scope and focus of our current research, which primarily revolves around assessing fruit edibility quality through a novel data model, not physiological analysis. Our study, as it stands, offers a new contribution to the field by applying a new data model for analyzing fruit quality, which to our knowledge, is the first study of its kind. The primary objective of our research is to establish a foundational understanding of fruit quality metrics using this model, which we believe provides a new method to compare fruit quality quantitatively, no matter in coating experiments, or other postharvest treatments. The use of Principal Component Analysis (PCA) in our study, while a common tool, is employed in a novel manner to interpret complex data sets related to fruit quality. We recognize the importance of conducting more detailed physiological and biochemical analyses, such as profiling phenolic compounds. However, we intended this research to lay the groundwork for such subsequent studies. As you correctly pointed out, there are numerous references indicating changes in basic analysis parameters (pH, acidity, soluble solids, firmness, color, etc.) and total phenols when fruits are coated. These studies provide an excellent backdrop against which our current research can be contextualized. Our findings, which align with these earlier studies, underscore the validity and reliability of our novel data model. Incorporating additional experiments to profile phenolic compounds and conduct specific physiological analyses would significantly expand the scope of our current research. While we agree that these are valuable areas of study, we believe they are more suited to a follow-up study. Our team is already in the process of designing this subsequent research, where we aim to delve deeper into the physiological and biochemical changes in coated fruits, building upon the foundation established by our current work. We acknowledge the importance of the analyses you have suggested, we respectfully submit that the current focus of our manuscript remains relevant and valuable to the field. Our novel approach to analyzing fruit quality through a new data model presents a unique contribution and sets the stage for more detailed future studies. We hope that this explanation clarifies the intent and scope of our research. We have addressed, and are open to further discussion of the manuscript to better clarify its scope and objectives.
Reviewer 3 Report
Comments and Suggestions for Authors
Accept
Author Response
Thank you!
Reviewer 4 Report
Comments and Suggestions for Authors
This manuscript can be published.